# Corporate Carbon Information Disclosure and Financing Costs: The Moderating Effect of Sustainable Development

**Guangyang Wang [1], Xinxuan Lou [2], Jianfei Shen [3,*], Erli Dan [3], Xinyuan Zheng [3], Jiaxin Shao [3] and Jingjie Li [3]**

[1] State Grid Materials Co., Ltd., Beijing 100120, China; danxiaojian@sina.cn
[2] State Grid Corporation of China, Beijing 100031, China; mayuzetg@163.com
[3] School of Economics and Management, North China Electric Power University, Beijing 102206, China; 120192106151@ncepu.edu.cn (E.D.); xyzhengdr@163.com (X.Z.); 120202206211@ncepu.edu.cn (J.S.); lijingjie@ncepu.edu.cn (J.L.)
[*] Correspondence: shenjianfei@263.net.cn

**Abstract:** With the Global Reporting Initiative (a provider of the global best practice for impact reporting) systematically helping parties to understand and exchange issues such as climate change and formulating authoritative sustainability reporting guidelines, corporate sustainable development is becoming more and more critical for companies. Moreover, corporate carbon information disclosure has the potential to promote corporate financing after the Green Climate Fund has been playing their part in climate finance. Previous studies focused more on the cost of equity. Considering the volatility of the capital market, the cost of equity financing is more unstable and complex. This study limited the financing cost to the cost of debt, took Chinese listed companies from 2009 to 2021 as a research sample, and explored the relationship between corporate carbon information disclosure, sustainable development, and financing costs. This study adopted fixed-effects (within) regression or random-effects GLS regression (defined through the Breusch and Pagan Lagrange multiplier test for random effects and the Hausman test) as estimation methods to control individual effects and endogenous problems brought by time. At the same time, the model was modified when there was heteroscedasticity and autocorrelation accordingly. The results show that the more carbon information disclosure, the lower the financing cost; sustainable development weakens the inhibitory effect of carbon information disclosure on financing costs. This study affirms the financing value of reducing information asymmetry, and found that sustainable development (internal growth capacity) may increase the cost of debt. The stronger the sustainable development is, the more financing needs may be, thus raising the cost of debt. This study not only implies that creditors may attach importance to the value of carbon information disclosure at the time of borrowing, but also provides theoretical evidence for the government or securities regulators to speed up the mandatory carbon information disclosure.

**Keywords:** corporate carbon information disclosure; sustainable development; financing cost



## 1. Introduction

### 1.1. Backgrounds

With the rapid development of the global economy, varieties of economic forms and the continuous innovation of economic financing models not only brought huge business opportunities, but also brought obstacles to the financial development of enterprises. Management and operation are further developed, and the sustainable growth rates provide a basis for managers to make strategic decisions [1]. In the broad sense, the sustainable development of enterprises is the coordinated development between enterprises and their internal and external environment. In the narrow sense, it can be understood as the balanced growth that an enterprise adapts to enterprise resources in the process of survival and sustainable development. Moreover, the sustainable development capability

of an enterprise is a necessary factor for the enterprise to proceed from small to large and expand from weak to strong [2]. The problem of "financing is difficult and expensive" has existed for a very long time in China's capital market, and it has hindered the daily and further development of many enterprises. Therefore, how to lower financing costs is an important issue in the development of enterprises. The sustainable development of enterprises has certain reference significance and it has become one of the criteria for evaluating the quality of enterprises [3]. Financial institutions such as banks indicated that they are more willing to lend to enterprises with high sustainable development than enterprises with low sustainable development [4]. However, most studies focus on the stakeholders aspect of corporate sustainable development and there is a lack of in-depth research on the influence of sustainable development in corporate financing. In a way, it is of great significance to investigate whether corporate sustainable development can affect the financing cost of enterprises.

In addition, governments have been actively responding to the "low-carbon economy". As the second largest carbon emitter in the world, China is particularly crucial to reducing carbon emission. Trials for carbon emission rights were piloted in China in 2007. Moreover, China launched carbon emission permits trade in 2017. As a means of demonstration of energy conservation and emission reduction, corporate carbon information disclosure has become one of the important bases of evaluating corporate credit risk, especially in the context of green credit and green supply chain management [5]. Since enterprises with more carbon information disclosure and better carbon performance have lower external environmental risks and require lower return on investment, carbon information has a certain impact on the reduction of financing costs.

From the perspective of corporate sustainable development, studying the relationship between corporate carbon information disclosure and corporate debt financing costs in the context of low-carbon economic development can help broaden the factors that affect debt financing costs and test the market value of information disclosure, which is beneficial to the promotion of carbon information disclosure of listed companies. Based on the sustainable growth model of American financial scientist Robert Higgins and the theory of information asymmetry, this study attempts to investigate the relationship between sustainable development and carbon information disclosure and the cost of financing. Main research questions: 1. Can corporate carbon information disclosure reduce the cost of corporate financing? 2. Will the sustainable development of enterprises reduce the cost of corporate financing? 3. Do sustainable development and carbon information disclosure have a synergistic effect on the cost of corporate financing?

*1.2. Literature Review*

Referring to literature review, the sustainable development of an enterprise may include the enterprise's awareness and ability of independent innovation, resource allocation efficiency, and management competencies [6]. The sustainable development of enterprises refers to the ability to achieve business goals and to maintain the advantages of enterprises, to continue to make profits in the process of long-term survival and development. From the view of corporate governance and financial management, it can be divided into strategic development capabilities, production and operation capabilities, profitability, solvency, sustainable growth capabilities, etc. Some scholars also use a single standard, such as the company's net profit growth level in the past three years, to measure the company's sustainable development capability [7].

Scholars have discussed the influencing factors of corporate financing costs from the perspective of both internal and external aspects, which include equity structure, corporate development (financial performance, etc.), corporate governance, internal control, managerial displacement, and so on [8]. The shareholding ratio of institutional investors has a certain effect on the financing cost of enterprises, and the financing cost of enterprises with multiple major shareholders or with stringent internal control is lower [9–11]. Previous studies have focused more on the role of corporate development on financing costs. There

is a negative correlation between the development capabilities (growth ability or financial performance) of an enterprise and its financing cost and the effect is more pronounced in a chaotic market environment and imperfect institutions [12–14]. In terms of financial indicators, financing cost is significantly related to enterprise scale and profitability, but not to industry characteristics and growth stages [15]. Managerial displacement will increase corporate financing costs, and this effect is more pronounced in state-owned enterprises [16]. From the view of the external perspective of enterprises, the national financial and monetary policy helps maintain the stability of the financial market and positively affects the financing cost of enterprises [17,18]. Other external factors include national political power, international capital flows, and the future growth environment (risk) of enterprises [19–21]. Although there are many studies on corporate financing costs, the studies on the relationship between corporate sustainable development and financing costs are still in the exploratory stage.

Before investigating carbon information disclosure and corporate financing costs, companies that actively undertake social responsibilities and invest in ESG will have lower financing costs [22–24]. Further, the more corporate ESG disclosure or the more information disclosure of social responsibility, the lower the corporate financing cost [25,26]. Considering the potential cost of corporate information disclosure, corporate social responsibility information disclosure may increase corporate financing costs to a certain extent [27]. The discrepancies in the above empirical results are significantly related to the scale of enterprises, industry characteristics, and types of environmental information (monetary environmental information, etc.) [28,29].

With the development of China's carbon trading market, the research on carbon information disclosure has become more and more in-depth. The relationships between carbon information disclosure and corporate value, and financial performance and capital cost were mainly explored. For example, high-quality carbon information disclosure is helpful for companies to control financial risks, and can effectively promote corporate performance in the current period as well as the next period [30,31]. The research on the capital cost of carbon information disclosure involves the cost of debt financing, the cost of equity financing, the roles of government regulation and environmental regulation, executive incentives, corporate carbon performance, and the nature of property rights. Specifically, for companies with poor carbon performance, their carbon information disclosure will significantly reduce corporate financing costs, whereas it is not significant for companies with better carbon performance [32,33]. Moreover, non-state-owned enterprises' disclosure of carbon information has a more obvious effect on reducing financing costs than state-owned enterprises [34,35]. Through the study of sample companies in the CDP report, it is concluded that companies that voluntarily choose to disclose carbon information will have better loan conditions in the case of information asymmetry and poor information transparency in the capital market [36]. Previous research mainly focused on the relationship between corporate social responsibility disclosure and capital cost, whereas lack of depth is a limiting factor of the studies on the relationship between carbon information disclosure and corporate financing cost.

From the perspective of green credit policy and corporate social responsibility, corporate social responsibility has a certain impact on the cost of corporate debt capital [37]. In addition, there is a significant correlation between corporate carbon information disclosure and debt financing costs [38,39]. At present, domestic and foreign scholars have conducted preliminary research on the relationship between each pair of corporate sustainable development, carbon information disclosure, and financing costs, but there is a lack of research on the triadic relations between sustainable development, carbon information disclosure, and corporate financing costs. Moreover, the proxy variable of carbon information disclosure is often too generalized, such as substituting it with environmental information disclosure or simply using dummy variables. In the context of severe climate change, based on the corporate sustainable development model and the theory of information asymmetry, to construct the link between corporate sustainable development, carbon in-

formation disclosure, and debt financing costs is of great value in the exploratory factors that influence debt financing costs and the formulation of information disclosure policies as well as demonstrating the advantages of corporate financing. Therefore, this paper uses the data of Chinese A-share listed companies from 2009 to 2021 as the research sample to explore the relationship between corporate sustainable development, carbon information disclosure, and corporate financing costs. The interactions predicted in the paper and to be tested in empirical analyses are illustrated in Figure 1.

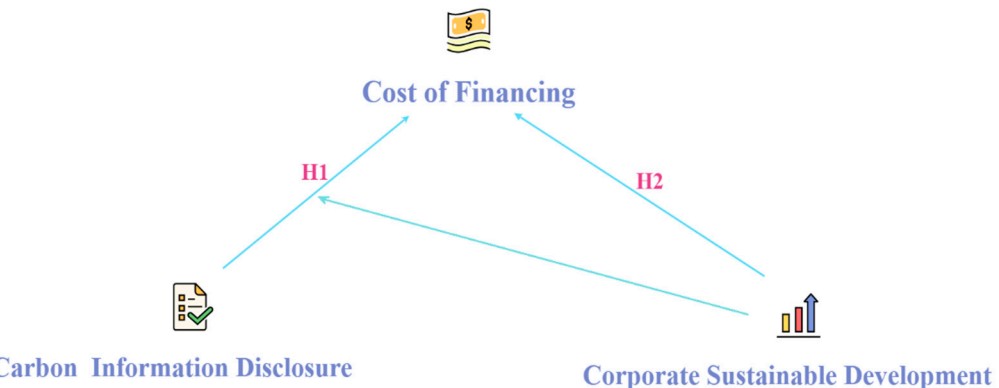

**Figure 1.** Interactions of sustainable development in the relationship between corporate carbon information disclosure and financing costs.

*1.3. Research Hypothesis*

Information asymmetry will cause adverse selection and moral hazard problems between enterprises and investors. Signaling theory suggests that public disclosure of corporate ESG performance can contribute to lowering financing costs by reducing information asymmetry [40]. Therefore, enterprises that actively disclose their carbon information will gain some competitive advantages to an extent. On the one hand, corporate carbon information disclosure can show the public that its business is in good condition and enhance investor confidence; on the other hand, companies can show the government their determination to actively respond to environmental protection and emission reduction policies and fulfill their social responsibilities. Since 2007, the China Banking Regulatory Commission has successively issued the Guiding Opinions on Credit Granting for Energy Conservation and Emission Reduction, Guidelines for Green Credit, Guidelines for Energy Efficiency Credit, and other guiding documents. At present, a green credit system framework has been basically established [41–43]. Various domestic financial institutions and commercial banks have also issued a series of green credit policies, which indicates that the credit standards for domestic enterprises in heavily polluting industries have been further raised, and it also urges enterprises to continuously improve their ESG disclosure, thereby improving their financing access capabilities. As the whole society pays more attention to environmental protection and government departments continue to strengthen environmental supervision of enterprises, external investors such as creditors have further increased requirements for enterprises to save energy and reduce emissions [44]. What's more, legitimacy theory predicts that companies can improve their carbon information disclosure, which may help reduce their own debt costs [32–36]. Therefore, hypothesis 1a and hypothesis 1b are proposed in this study.

**H1a.** *Corporate carbon information disclosure lowers the cost of debt financing.*

**H1b.** *Corporate carbon information disclosure increases the cost of debt financing.*

According to Higgins' sustainable growth rates model, the growth rate of enterprises is affected by corporate financing policies. The sustainable development of enterprises affects financing costs by reflecting profitability and retained earnings. Starting from the

stakeholder theory, the production and operation of an enterprise is closely related to its shareholders and creditors. In addition, enterprises must pay attention to factors such as environmental governance, business operation, and social responsibility, and consider the interests of multiple parties to reduce the negative evaluation of enterprises from the outside world, thereby affecting enterprise valuation and enterprise financing costs [45,46]. From the perspective of fulfilling social responsibilities, corporate carbon information disclosure is an important act to fulfill social responsibilities and establish a good image. Consumers will be more willing to buy the products of companies with a strong sense of social responsibility, thereby improving corporate operating efficiency. When investors make investment evaluations, they also have strong investment confidence in the company because of the company's active disclosure behavior, which may reduce the cost of corporate debt financing [47]. The more carbon information disclosure with corporate sustainable development, the lower the cost of debt financing. Therefore, we proposed assumptions H2 and H3.

**H2a.** *Corporate sustainable development positively affects the cost of debt financing.*

**H2b.** *Corporate sustainable development negatively affects the cost of debt financing.*

**H3a.** *Corporate sustainability strengthens the relationship between corporate carbon information disclosure and debt costs.*

**H3b.** *Corporate sustainability weakens the relationship between corporate carbon information disclosure and debt costs.*

The sections of this article are as follows: the second chapter, the literature review part, systematically analyzes the progress of the main variables of this study and the relationship between them, which provides a basis for the construction of the follow-up research framework; the third chapter, the research design part, details the sample selection, data sources, and research hypotheses, explains variables, and constructs the empirical model; the fourth chapter is the empirical results part, including descriptive statistics, correlation analysis, regression analysis, robustness test, etc.; and the fifth chapter is the conclusion and research prospect.

The results show that the more carbon information disclosure, the lower the financing cost; sustainable development weakened the inhibitory effect of carbon information disclosure on financing costs. On the basis of ensuring the correlation, data reliability, and authenticity of the proxy variables of carbon information disclosure, this study verified the advantages of carbon information disclosure in corporate financing, and affirmed the value of reducing information asymmetry in financing. This study not only implies that improving corporate voluntary carbon information disclosure will contribute to corporate financing, but also provides theoretical evidence for the government or securities regulators to speed up the mandatory carbon information disclosure.

## 2. Materials and Methods

### 2.1. Data Sources

Companies' financial data came from the RESSET database and annual reports, and carbon information disclosure data were collected from social responsibility reports, sustainable development reports, or ESG reports. This study selected reports that disclosed corporate carbon emissions or carbon emission reductions as samples. The sample covered 26 industries such as electricity, heat production and supply, ferrous metal smelting and rolling, non-metallic mineral products, coal mining and washing, chemical raw materials and chemical products manufacturing, non-ferrous metal smelting and rolling, gas production and supply industry, food manufacturing, oil and gas extraction, wine, beverage, and refined tea manufacturing, ferrous metal mining and dressing, general equipment

manufacturing, pharmaceutical manufacturing, metal products, automobile manufacturing, non-ferrous metal mining industry selection, electrical machinery and equipment manufacturing, instrumentation manufacturing, computer, communication, and other electronic equipment manufacturing, water production and supply, finance, petroleum processing, coking and nuclear fuel processing, chemical fiber manufacturing, special equipment manufacturing, railways, ships, aerospace and other transportation equipment manufacturing, and textile and apparel industries. There were 474 observations. In addition, 99% of quantiles were abbreviated and normalized for the main variables, and explanatory and moderator variables were centered before constructing interaction terms.

*2.2. Variables*

The financing cost (explained variable) reflects the interest cost of the enterprise obtaining funds from the bank or other external financing channels. Referring to the existing literature, it is common to use the debt financing cost of enterprises as a proxy variable of financing cost [21,48,49]. Considering that the cost of equity financing has a more complex impact mechanism than the cost of debt financing, this study takes the cost of debt financing (DebtCost) as an important component of financing cost, and measures the cost of debt financing with the ratio of interest expenses and interest-bearing debt.

The main representative point of the theory of sustainable growth is from Professor Higgins, who proposed that the sustainable growth rate is the largest sales that an enterprise can achieve without issuing new shares or changing operating efficiency and financial policies. The growth rate can more accurately measure the sustainable development of enterprises. Thus, this study chooses the sustainable growth rate (SustaDe) as the explanatory variable of the sustainable development of enterprises.

Carbon information disclosure (CarbDisc) is the behavior of companies to provide quantitative or qualitative carbon information such as monetary value to readers. The quality of carbon information disclosure can be measured by the number of carbon information disclosure items. This study collected carbon information items in corporate social responsibility reports, sustainability reports, and environmental, social, and governance reports. In total, 62 items were collected, including carbon emissions, carbon emission reductions, identification of climate-related risks, emission reduction targets, carbon assets management, low-carbon transfer plans, low-carbon issues in the value chain, green and low-carbon office measures, and so on.

This paper also divided the samples into subsamples of high-carbon-intensive industries and low-carbon-intensive industries according to the carbon intensity divided by researchers. Due to the large difference between the two characteristics, it may be truer to study the difference between the carbon information volume or carbon emission data information on enterprises in the two industries.

In order to more comprehensively examine the correlation between corporate sustainable development, carbon information disclosure, and financing costs, considering data availability, this paper also considered other factors that have an important impact on corporate financing costs as control variables. The company size of listed companies can reflect the company's disposable resources to a certain extent. Generally, larger companies have more capital accumulation, stronger sustainable development capabilities, higher carbon information disclosure, and lower financing costs [50]. This study chose assets (Asset) and liabilities (Liab) as proxy variables of company size. The higher the profitability of an enterprise, the lower the financing cost [51]. Two indicators, operating income (Sales) and return on total assets (ROA), were selected as proxy variables for corporate profitability. From the perspective of solvency, capital utilization and operation, quick ratio (QuickRa), fixed asset ratio (FixedRa), and total asset turnover ratio (TotalTu) were selected as control variables [52]. Equity concentration is related to corporate governance and corporate decision-making and is an important indicator to measure the status of corporate equity distribution, corporate stability, and corporate structure, and may have an impact on financing costs [53]. This study set equity concentration as a control variable,

and accordingly took equity concentration 1 (EC1), equity concentration 5 (EC5), equity concentration 10 (EC10), and equity concentration 11 (EC11) as proxy variables. At the same time, the control variables also included the proportion of state-owned shares (StateSP), the proportion of legal person shares (LePSP), the proportion of tradable A-shares (TrdaSP), and the proportion of tradable H-shares (TrdhSP). The specific descriptions of all the variables involved in this model are shown in Table 1.

**Table 1.** Definition of Variables.

| Variables | Name | Proxy Variables | Code | Variable Description |
|---|---|---|---|---|
| Explained Variable | Financing cost | Debt financing cost | DebtCost | the ratio of interest expenses and interest-bearing debt |
| Explanatory Variables | Carbon information disclosure | Quality of carbon information disclosure | CarbDisc | the number of carbon information disclosure items |
| | Sustainable development | Sustainable growth rate | SustaDe | (current net profit/beginning shareholders' equity) × current earnings retention rate × 100% |
| Control variables | Carbon intensity | Carbon-intensive industries | Indus | dummy variable (0 = low carbon, 1 = carbon-intensive industries) |
| | Size | Asset | Asset | |
| | | Liabilities | Liab | |
| | Profitability (Profit) | Operating income | Sales | |
| | | Return on total assets | ROA | |
| | Solvency (Solven) | Quick ratio | QuickRa | |
| | Capital utilization (CapiU) | Fixed asset ratio | FixedRa | |
| | | Total asset turnover ratio | TotalTu | |
| | Equity concentration (EquityConcen) | Shareholding ratio of the first largest shareholder | EC1 | |
| | | Shareholding ratio of top 5 shareholders | EC5 | |
| | | Shareholding ratio of top 10 shareholders | EC10 | |
| | | Number of shareholders | EC11 | Natural logarithm of the number of shareholders at the end of the period |
| | Stock liquidity (StockLiq) | Proportion of tradable A-shares | TrdaSP | RMB ordinary stocks |
| | | Proportion of tradable H-shares | TrdhSP | State-owned shares listed in Hong Kong |

### 2.3. Research Model

In order to solve the problem of missing variables, a panel data model was used for regression. In regression, heteroscedasticity, autocorrelation, and other issues were fully considered, and FGLS estimation was used to revise the regression coefficient accordingly. This study fully considered the cross-section, time series, and individual characteristics to make the regression results more effective, and established the following models:

Model 1:

$$DebtCost_{i,t} = \alpha \quad + \beta_1 CarbDisc_{i,t} + \beta_2 SustaDe_{i,t} + \beta_3 Indus_{i,t} + \beta_4 Size_{i,t} + \beta_5 \text{Profit}_{i,t}$$
$$+ \beta_6 \text{Solven}_{i,t} + \beta_7 \text{CapiU}_{i,t} + \beta_8 \text{EquityConcen}_{i,t} + \beta_9 \text{StockLiq}_{i,t} + v_i + \varepsilon_{it}$$

Model 2:

$$DebtCost_{i,t} = \alpha + \quad \beta_1 CarbDisc_{i,t} + \beta_2 SustaDe_{i,t} + \beta_3 CarbDisc_{i,t} * SustaDe_{i,t} + \beta_4 Indus_{i,t}$$
$$+ \beta_5 Size_{i,t} + \beta_6 \text{Profit}_{i,t} + \beta_7 \text{Solven}_{i,t} + \beta_8 \text{CapiU}_{i,t} + \beta_9 \text{EquityConcen}_{i,t}$$
$$+ \beta_{10} \text{StockLiq}_{i,t} + v_i + \varepsilon_{it}$$

where $DebtCost_{i,t}$ is debt financing cost and $SustaDe_{i,t}$ and $CarbDisc_{i,t}$ represent sustainable development and carbon information disclosure, respectively. $CarbDisc * SustaDe_{i,t}$ represents the interaction term of sustainable development and carbon information disclosure. Control variables included company size, profitability, solvency, capital utilization, equity concentration, and stock liquidity, and $v_i + \varepsilon_{it}$ is the disturbance term.

## 3. Results

### 3.1. Descriptive Statistics

This study collected the social responsibility reports, environmental reports, sustainable development reports, and ESG reports of listed companies from 2009 to 2021. The result was 97 valid companies and 474 observed values. As shown in Table 2, the number of observable values increased from 2 in 2009 to 65 in 2021 with an increasing trend in each year.

**Table 2.** Annual sample description.

| Year | Freq. | Percent | Cum. |
|---|---|---|---|
| 2009 | 2 | 0.42 | 0.42 |
| 2010 | 8 | 1.69 | 2.11 |
| 2011 | 13 | 2.74 | 4.85 |
| 2012 | 18 | 3.80 | 8.65 |
| 2013 | 20 | 4.22 | 12.87 |
| 2014 | 30 | 6.33 | 19.20 |
| 2015 | 36 | 7.59 | 26.79 |
| 2016 | 45 | 9.49 | 36.29 |
| 2017 | 65 | 13.71 | 50.00 |
| 2018 | 52 | 10.97 | 60.97 |
| 2019 | 60 | 12.66 | 73.63 |
| 2020 | 60 | 12.66 | 86.29 |
| 2021 | 65 | 13.71 | 100.00 |
| Total | 474 | 100.00 | |

The descriptive statistics of the variables are shown in Table 3. The independent variable SustaDe had a minimum and maximum value of −0.251 and 0.366, a mean value of 0.079, and a standard deviation of 0.087. The findings indicate that there is a large difference in sustainable development among enterprises. The mean value of CarbDisc for independent variables was 8.344, the lowest and highest amounts were 0 and 31, and the standard deviation was 5.560, indicating that the current level of carbon information disclosure of enterprises is uneven. China should actively publicize and promote the sustainable development of enterprises and carbon information disclosure, promoting the voluntary participation of enterprises in formulating sustainable development strategies and carbon information disclosure. The dependent variable DebtCost had minimum and maximum values of 0.009 and 0.341, a mean value of 0.055, and a standard deviation of 0.046. The deviation of the mean value of debt financing cost in the sample companies was large, which may be due to the large number of industry types of the sample companies.

The mean values of the companies' operating revenue (Sales) and return on total assets (ROA) were 1129.291 and 0.043, respectively, indicating that the sample enterprises

had high profitabilities. The mean value of company size (Asset) was 6938.593, and the standard deviation was 27,952.573, which means there was a large difference between sample enterprise sizes. The first largest shareholders' shareholding ratio (EC1) had a mean value of 0.423 with a high dispersion. The ownership of some samples was relatively concentrated. The sum of the shareholding ratios of the top five shareholders (EC5) and the top ten shareholders (E11) were similar to the descriptive statistical results of the shareholding ratio of the first largest shareholder (EC1). To sum up, most of the sample data had little volatility, low dispersion, and good stability, and the sample selection was reasonable and representative.

**Table 3.** Sample descriptive statistics.

| Variable | Obs | Mean | Std. Dev. | Min | Max |
|---|---|---|---|---|---|
| DebtCost | 474 | 0.055 | 0.046 | 0.009 | 0.341 |
| SustaDe | 474 | 0.079 | 0.087 | −0.251 | 0.366 |
| CarbDisc | 474 | 8.344 | 5.560 | 0 | 31 |
| Sales | 474 | 1129.291 | 3327.139 | 3.670 | 29,661.93 |
| Asset | 474 | 6938.539 | 27,952.573 | 18.187 | 302,539.81 |
| Liab | 474 | 5932.202 | 25,591.827 | 5.013 | 276,398.59 |
| ROA | 474 | 0.043 | 0.048 | −0.200 | 0.281 |
| FixedRa | 474 | 0.295 | 0.215 | 0.001 | 0.876 |
| QuickRa | 474 | 1.048 | 0.749 | 0.085 | 7.958 |
| TotalTu | 474 | 0.585 | 0.442 | 0.023 | 2.561 |
| EC1 | 474 | 0.423 | 0.175 | 0.078 | 0.990 |
| EC5 | 474 | 0.659 | 0.174 | 0.232 | 1.005 |
| EC10 | 474 | 0.699 | 0.162 | 0.278 | 1.012 |
| EC11 | 474 | 11.453 | 1.268 | 0.693 | 14.054 |
| TrdaSP | 474 | 0.861 | 0.182 | 0 | 1.000 |
| TrdhSP | 474 | 0.125 | 0.167 | 0 | 0.962 |

*3.2. Result of Correlation Coefficients*

Table 4 shows the correlation values among the variables. The correlation coefficient between sustainable development and financing cost was −0.180 with a significant negative correlation at the 1% level, indicating that improvement of sustainable development can promote financing cost reduction. There was no significant correlation between carbon information disclosure and financing cost. Financing cost was significantly negatively correlated with ROA and TotalTu, whereas it was significantly positively correlated with QuickRa at the 1% level. The correlation coefficients of EC1, EC5, EC10, and EC11 were 0.736, 0.693, and −0.222, respectively, which were significant at the 1% level.

**Table 4.** Correlation coefficients result.

| Variables | (1) | (2) | (3) | (4) | (5) | (6) | (7) | (8) | (9) | (10) | (11) | (12) | (13) | (14) | (15) | (16) |
|---|---|---|---|---|---|---|---|---|---|---|---|---|---|---|---|---|
| (1) DebtCost | 1.000 | | | | | | | | | | | | | | | |
| (2) SustaDe | −0.180 ***<br>(0.000) | 1.000 | | | | | | | | | | | | | | |
| (3) CarbDisc | −0.006<br>(0.897) | −0.006<br>(0.895) | 1.000 | | | | | | | | | | | | | |
| (4) Sales | 0.062<br>(0.177) | 0.011<br>(0.804) | 0.444 ***<br>(0.000) | 1.000 | | | | | | | | | | | | |
| (5) Asset | −0.028<br>(0.543) | 0.044<br>(0.340) | 0.334 ***<br>(0.000) | 0.330 ***<br>(0.000) | 1.000 | | | | | | | | | | | |
| (6) Liab | −0.029<br>(0.534) | 0.045<br>(0.331) | 0.318 ***<br>(0.000) | 0.299 ***<br>(0.000) | 0.999 ***<br>(0.000) | 1.000 | | | | | | | | | | |
| (7) ROA | −0.099 **<br>(0.030) | 0.660 ***<br>(0.000) | 0.002<br>(0.958) | −0.031<br>(0.496) | −0.134 ***<br>(0.003) | −0.138 ***<br>(0.003) | 1.000 | | | | | | | | | |
| (8) FixedRa | −0.097 **<br>(0.035) | −0.075<br>(0.103) | −0.305 ***<br>(0.000) | −0.091 **<br>(0.047) | −0.271 ***<br>(0.000) | −0.274 ***<br>(0.000) | 0.095 **<br>(0.038) | 1.000 | | | | | | | | |
| (9) QuickRa | 0.133 ***<br>(0.004) | 0.067<br>(0.145) | 0.037<br>(0.426) | −0.069<br>(0.133) | 0.027<br>(0.558) | 0.027<br>(0.555) | 0.305 ***<br>(0.000) | −0.304 ***<br>(0.000) | 1.000 | | | | | | | |
| (10) TotalTu | −0.089 *<br>(0.053) | 0.173 ***<br>(0.000) | 0.202 ***<br>(0.000) | 0.224 ***<br>(0.000) | −0.234 ***<br>(0.000) | −0.241 ***<br>(0.000) | 0.365 ***<br>(0.000) | −0.049<br>(0.288) | −0.004<br>(0.932) | 1.000 | | | | | | |
| (11) EC 1 | −0.112 **<br>(0.015) | 0.023<br>(0.610) | 0.137 ***<br>(0.003) | 0.189 ***<br>(0.000) | 0.075<br>(0.104) | 0.065<br>(0.160) | 0.009<br>(0.850) | 0.193 ***<br>(0.000) | −0.041<br>(0.372) | 0.071<br>(0.123) | 1.000 | | | | | |
| (12) EC 5 | −0.095 **<br>(0.038) | −0.022<br>(0.637) | 0.266 ***<br>(0.000) | 0.214 ***<br>(0.000) | 0.172 ***<br>(0.000) | 0.162 ***<br>(0.000) | −0.015<br>(0.748) | 0.083 *<br>(0.072) | −0.036<br>(0.431) | 0.046<br>(0.314) | 0.736 ***<br>(0.000) | 1.000 | | | | |
| (13) EC 10 | −0.076 *<br>(0.099) | −0.022<br>(0.635) | 0.298 ***<br>(0.000) | 0.224 ***<br>(0.000) | 0.196 ***<br>(0.000) | 0.186 ***<br>(0.000) | −0.026<br>(0.568) | 0.063<br>(0.174) | −0.026<br>(0.568) | 0.015<br>(0.739) | 0.693 ***<br>(0.000) | 0.985 ***<br>(0.000) | 1.000 | | | |
| (14) EC 11 | 0.061<br>(0.186) | −0.049<br>(0.292) | 0.250 ***<br>(0.000) | 0.299 ***<br>(0.000) | 0.229 ***<br>(0.000) | 0.218 ***<br>(0.000) | −0.054<br>(0.238) | −0.080 *<br>(0.084) | −0.031<br>(0.506) | −0.154 ***<br>(0.001) | −0.222 ***<br>(0.000) | −0.161 ***<br>(0.000) | −0.143 ***<br>(0.002) | 1.000 | | |
| (15) TrdaSP | −0.022<br>(0.639) | −0.074<br>(0.108) | −0.426 ***<br>(0.000) | −0.216 ***<br>(0.000) | −0.480 ***<br>(0.000) | −0.477 ***<br>(0.000) | 0.085 *<br>(0.066) | 0.345 ***<br>(0.000) | −0.046<br>(0.322) | 0.008<br>(0.867) | −0.112 **<br>(0.015) | −0.389 ***<br>(0.000) | −0.397 ***<br>(0.000) | 0.106 **<br>(0.021) | 1.000 | |
| (16) TrdhSP | 0.043<br>(0.352) | 0.028<br>(0.541) | 0.454 ***<br>(0.000) | 0.256 ***<br>(0.000) | 0.540 ***<br>(0.000) | 0.536 ***<br>(0.000) | −0.108 **<br>(0.019) | −0.361 ***<br>(0.000) | 0.072<br>(0.118) | −0.082 *<br>(0.075) | 0.022<br>(0.628) | 0.394 ***<br>(0.000) | 0.409 ***<br>(0.000) | 0.207 ***<br>(0.000) | −0.865 ***<br>(0.000) | 1.000 |

*** $p < 0.01$, ** $p < 0.05$, * $p < 0.1$.

*3.3. Regression Analyses*

In this study, fixed-effect or random-effect models were applied. Variables were omitted if there was collinearity. We used the Breusch and Pagan Lagrangian multiplier test for random effects and the Hausman test to determine which of the above two models was more suitable. Then the corresponding heteroscedasticity test (such as the modified Wald test for groupwise heteroskedasticity in the fixed-effect regression model) and Wooldridge test for autocorrelation in panel data were carried out. Finally, the generalized least-square FGLS estimation was used to modify the model. STATA, a statistical software, was used.

The results from the Breusch and Pagan Lagrangian multiplier test for random effects and the Hausman test were chibar2(01) = 208.59, chi2(16) = 49.87 significantly, as shown in Table 5. It showed that the fixed-effect model was more suitable. The modified Wald test for groupwise heteroskedasticity in the fixed-effect regression model indicated that there was heteroscedasticity(chi2(16) = $5.8 \times 10^{31}$). The result from the Wooldridge test for autocorrelation in panel data indicated that there was autocorrelation (F (1, 66) = 15.754, $p = 0.0002$). After being modified, the FGLS results show that the more carbon information was disclosed, the lower the cost of debt. However, the impact of sustainable development on debt costs was not significant.

**Table 5.** Regression result.

| | Model 1 | | |
|---|---|---|---|
| | **(1) FE** | **(2) RE** | **(3) FGLS** |
| **VARIABLES** | | **DebtCost** | |
| CarbDisc | −0.103 ** (−2.46) | −0.129 *** (−3.31) | −0.065 *** (−2.58) |
| SustaDe | 0.014 (0.53) | 0.018 (0.70) | 0.020 (1.59) |
| Indus | 0.051 (0.13) | −0.237 (−1.63) | −0.127 ** (−2.16) |
| Sales | 0.659 *** (5.52) | 0.313 *** (3.53) | 0.104 ** (2.17) |
| Asset | −0.816 *** (−3.26) | −0.384 * (−1.87) | −0.115 (−1.00) |
| Liab | 0.012 (0.06) | 0.046 (0.28) | −0.046 (−0.49) |
| ROA | −0.361 (−0.43) | −0.349 (−0.44) | −0.307 (−0.57) |
| FixedRa | 1.189 *** (3.89) | 0.873 *** (3.40) | 0.513 *** (3.79) |
| QuickRa | 0.086 (1.29) | 0.088 (1.57) | 0.050 (1.27) |
| TotalTu | −1.039 *** (−4.68) | −0.681 *** (−3.77) | −0.328 *** (−2.78) |
| EC1 | 0.265 (0.51) | 0.167 (0.43) | −0.070 (−0.40) |
| EC5 | −1.392 (−0.99) | −1.814 (−1.46) | −1.774 *** (−2.99) |
| EC10 | 0.769 (0.59) | 1.072 (0.89) | 1.748 *** (2.88) |
| EC11 | −0.000 (−0.00) | −0.020 (−0.43) | 0.057 ** (2.11) |
| TrdaSP | −0.001 (−0.10) | 0.001 (0.19) | −0.003 (−0.85) |
| TrdhSP | 0.014 (1.33) | 0.010 * (1.68) | −0.000 (−0.02) |
| Constant | −0.971 (−1.33) | −1.777 *** (−4.35) | −2.959 *** (−11.57) |
| F | 5.40 *** | - | - |
| chi2 | - | 64.12 *** | 126.43 *** |
| Observations | 471 | 471 | 460 |
| R-squared | 0.193 | 0.162 | - |
| Number of idcode | 95 | 95 | 84 |
| idcode FE | YES | YES | YES |
| Year FE | YES | YES | YES |

*t*-statistics or *z*-statistics in parentheses. *** $p < 0.01$, ** $p < 0.05$, * $p < 0.1$.

In the process of moderating the effect test, the fixed model was more suitable. The statistical results are shown in Table 6. The results show that the more carbon information was disclosed, the lower the cost of debt. In addition, sustainable development weakened the inhibitory effect of carbon information disclosure on the cost of debt.

**Table 6.** Regression result for moderating effect.

| VARIABLES | Model 2 | | |
|---|---|---|---|
| | **(1) FE** | **(2) RE** | **(3) FGLS** |
| | | **DebtCost** | |
| CarbDisc | −0.094 ** (−2.22) | −0.124 *** (−3.16) | −0.061 ** (−2.43) |
| SustaDe | −0.043 (−0.86) | −0.023 (−0.47) | −0.033 (−1.08) |
| CarbDisc * SustaDe | 0.030 (1.31) | 0.021 (0.97) | 0.023 * (1.85) |
| Indus | 0.037 (0.10) | −0.237 (−1.62) | −0.134 ** (−2.32) |
| Sales | 0.678 *** (5.64) | 0.324 *** (3.63) | 0.112 ** (2.32) |
| Asset | −0.878 *** (−3.45) | −0.412 ** (−1.99) | −0.131 (−1.15) |
| Liab | 0.037 (0.20) | 0.059 (0.36) | −0.037 (−0.40) |
| ROA | −0.089 (−0.10) | −0.183 (−0.23) | 0.014 (0.03) |
| FixedRa | 1.146 *** (3.73) | 0.864 *** (3.35) | 0.523 *** (3.96) |
| QuickRa | 0.085 (1.28) | 0.087 (1.56) | 0.048 (1.24) |
| TotalTu | −1.102 *** (−4.85) | −0.710 *** (−3.88) | −0.345 *** (−2.93) |
| EC1 | 0.199 (0.39) | 0.157 (0.40) | −0.095 (−0.56) |
| EC5 | −1.347 (−0.96) | −1.831 (−1.48) | −1.705 *** (−2.98) |
| EC10 | 0.763 (0.59) | 1.113 (0.93) | 1.713 *** (2.92) |
| EC11 | −0.005 (−0.09) | −0.021 (−0.45) | 0.058 ** (2.16) |
| TrdaSP | −0.000 (−0.01) | 0.001 (0.20) | −0.003 (−0.92) |
| TrdhSP | 0.014 (1.37) | 0.010 * (1.71) | −0.000 (−0.12) |
| Constant | −0.961 (−1.32) | −1.866 *** (−4.44) | −3.117 *** (−11.66) |
| F | 5.19 *** | - | - |
| chi2 | - | 65.27 *** | - |
| Observations | 471 | 471 | 460 |
| R-squared | 0.197 | 0.165 | - |
| Number of idcode | 95 | 95 | 84 |
| idcode FE | YES | YES | YES |
| Year FE | YES | YES | YES |

*** $p < 0.01$, ** $p < 0.05$, * $p < 0.1$.

### 3.4. Robustness Check

In view of the conclusion that the financial carbon information disclosure of enterprises in China's heavily polluting industries can reduce debt financing cost, and the non-financial carbon information disclosure cannot significantly reduce the cost of equity financing [34], this study adopted to replace independent variables to test the robustness. That is, replacing the number of carbon information disclosure items with the weighted carbon information disclosure (CarbDisc_). Specifically, different types of carbon information items were distinguished, and higher scores were assigned to the carbon information disclosure items that contained monetary or quantitative information. More specifically, 3 points, 2 points, and 1 point were for each monetary, quantitative, and qualitative carbon information item, respectively. The test showed that the fixed model was applicable. After correcting heteroscedasticity and autocorrelation, the results show that the more carbon information was disclosed, the lower the cost of debt, as shown in Table 7. In addition, sustainable development weakened the inhibitory effect of carbon information disclosure on the cost of debt, consistent with the original results.

**Table 7.** Regression result for Robustness Check.

| VARIABLES | Model 1 | | | Model 2 | | |
|---|---|---|---|---|---|---|
| | **(1) FE** | **(2) RE** | **(3) FGLS** | **(1) FE** | **(2) RE** | **(3) FGLS** |
| | | | **DebtCost** | | | |
| CarbDisc | −0.131 *** | −0.148 *** | −0.083 *** | −0.123 *** | −0.143 *** | −0.081 *** |
| | (−2.97) | (−3.57) | (−2.99) | (−2.74) | (−3.42) | (−2.94) |
| SustaDe | 0.012 | 0.016 | 0.020 * | −0.050 | −0.030 | −0.043 |
| | (0.47) | (0.63) | (1.66) | (−0.82) | (−0.52) | (−1.17) |

**Table 7.** *Cont.*

| | Model 1 | | | Model 2 | | |
|---|---|---|---|---|---|---|
| | **(1) FE** | **(2) RE** | **(3) FGLS** | **(1) FE** | **(2) RE** | **(3) FGLS** |
| **VARIABLES** | | | **DebtCost** | | | |
| CarbDisc * SustaDe | - | - | - | 0.027 | 0.020 | 0.024* |
| | | | | (1.12) | (0.87) | (1.78) |
| Indus | 0.027 | −0.226 | −0.120 ** | 0.014 | −0.225 | −0.127 ** |
| | (0.07) | (−1.55) | (−2.08) | (0.04) | (−1.54) | (−2.25) |
| Sales | 0.670 *** | 0.317 *** | 0.103 ** | 0.686 *** | 0.327 *** | 0.111 ** |
| | (5.63) | (3.58) | (2.13) | (5.72) | (3.66) | (2.30) |
| Asset | −0.795 *** | −0.377 * | −0.107 | −0.844 *** | −0.400 * | −0.124 |
| | (−3.19) | (−1.84) | (−0.95) | (−3.34) | (−1.94) | (−1.10) |
| Liab | −0.009 | 0.040 | −0.049 | 0.010 | 0.051 | −0.038 |
| | (−0.05) | (0.25) | (−0.53) | (0.06) | (0.31) | (−0.42) |
| ROA | −0.344 | −0.334 | −0.307 | −0.109 | −0.180 | 0.032 |
| | (−0.41) | (−0.42) | (−0.57) | (−0.13) | (−0.22) | (0.06) |
| FixedRa | 1.191 *** | 0.871 *** | 0.508 *** | 1.149 *** | 0.859 *** | 0.522 *** |
| | (3.92) | (3.40) | (3.78) | (3.75) | (3.34) | (4.02) |
| QuickRa | 0.085 | 0.088 | 0.047 | 0.084 | 0.087 | 0.045 |
| | (1.28) | (1.58) | (1.20) | (1.27) | (1.57) | (1.15) |
| TotalTu | −1.048 *** | −0.683 *** | −0.324 *** | −1.098 *** | −0.708 *** | −0.343 *** |
| | (−4.74) | (−3.78) | (−2.74) | (−4.87) | (−3.88) | (−2.90) |
| EC1 | 0.286 | 0.174 | −0.061 | 0.234 | 0.167 | −0.093 |
| | (0.56) | (0.44) | (−0.35) | (0.45) | (0.43) | (−0.56) |
| EC5 | −1.500 | −1.863 | −1.768 *** | −1.456 | −1.869 | −1.643 *** |
| | (−1.07) | (−1.50) | (−2.99) | (−1.04) | (−1.51) | (−2.91) |
| EC10 | 0.790 | 1.072 | 1.702 *** | 0.762 | 1.092 | 1.612 *** |
| | (0.61) | (0.90) | (2.82) | (0.59) | (0.91) | (2.77) |
| EC11 | −0.003 | −0.023 | 0.056 ** | −0.010 | −0.025 | 0.056 ** |
| | (−0.06) | (−0.49) | (2.07) | (−0.18) | (−0.53) | (2.08) |
| TrdaSP | −0.000 | 0.001 | −0.002 | 0.000 | 0.001 | −0.002 |
| | (−0.07) | (0.23) | (−0.82) | (0.03) | (0.27) | (−0.86) |
| TrdhSP | 0.014 | 0.011 * | 0.001 | 0.014 | 0.011 * | 0.000 |
| | (1.34) | (1.80) | (0.21) | (1.39) | (1.84) | (0.13) |
| Constant | −0.868 | −1.699 *** | −2.892 *** | −0.891 | −1.804 *** | −3.076 *** |
| | (−1.20) | (−4.15) | (−11.15) | (−1.23) | (−4.21) | (−11.13) |
| F | 5.61 *** | - | - | 5.35 *** | - | - |
| chi2 | - | 66.25 *** | 130.22 *** | - | 67.17 *** | 141.74 *** |
| Observations | 471 | 471 | 460 | 471 | 471 | 460 |
| R-squared | 0.199 | 0.168 | | 0.202 | 0.170 | |
| Number of idcode | 95 | 95 | 84 | 95 | 95 | 84 |
| idcode FE | YES | YES | YES | YES | YES | YES |
| Year FE | YES | YES | YES | YES | YES | YES |

*** $p < 0.01$, ** $p < 0.05$, * $p < 0.1$.

## 4. Discussion

The empirical results show that the more carbon information is disclosed, the lower the cost of debt. It is in line with the expectation of information asymmetry theory and legitimacy theory as well as some empirical verification about carbon profile and financing [54]. Moreover, sustainable development weakens the inhibitory effect of carbon information disclosure on the cost of debt. The stronger the sustainable development (internal growth capacity), the more financing demand may be, which increases the debt cost, thus showing a regulatory effect (weakening effect) in the inhibition relationship between carbon information disclosure and debt cost. In addition, the higher the carbon intensity of enterprises (carbon-intensive industries), the lower the cost of debt. This may be similar to the large-scale and long-term development of enterprises in carbon-intensive industries, and investors are still more interested in enterprises in carbon-intensive industries. Moreover,

the industry itself is covered by subsidy policies and financial support from other enterprises, that is, it has some preferential debts. The higher the fixed asset ratio or the slower the turnover, the higher the debt cost. From the perspective of corporate governance, the higher the share proportion of the top five shareholders, the lower the debt cost. From the perspective of stock circulation, the higher the number of shareholders (natural logarithm), the higher the debt cost. Relevant explanations are as follows: stock market development may lower the cost of equity [55]; the more shareholders, the more complex the stock circulation may be, resulting in greater debt financing constraints and higher debt costs; and what's more, the impact of circulating H-shares on the cost of debt is not significant.

## 5. Conclusions

This paper adopted Chinese listed companies from 2009 to 2021 as a sample, and explored the relationship between corporate sustainable development, carbon information disclosure, and debt costs based on the carbon information disclosure in their ESG reports, social responsibility reports, and sustainability reports. Adoption for fixed-effects (within) regression or random-effects GLS regression were identified as appropriate estimation methods after the Breusch and Pagan Lagrange multiplier test for random effects and the Hausman test. Then, the model was modified when there was heteroscedasticity and autocorrelation accordingly. The results show that the more carbon information disclosure, the lower the financing cost; sustainable development weakens the inhibitory effect of carbon information disclosure on financing costs. This study affirms the financing value of reducing information asymmetry, and found that sustainable development (internal growth capacity) may increase the cost of debt. The stronger the sustainable development is, the more financing needs may be, thus raising the cost of debt. The empirical results will help companies understand the value of carbon information disclosure in reducing debt financing costs. In addition, enterprises actively participating in carbon information disclosure will contribute to the realization of the national carbon emission reduction target to a great extent. Moreover, on the basis of available data, it is also of great value in exploring whether the relationship between sustainable development, carbon information disclosure, and debt financing costs is different in high-carbon industries or low-carbon industries.

In consideration of the distribution of sample observations (nearly half were concentrated in 2018–2021), the relatively high debt financing cost at the time of listing, and the decline of debt cost under the COVID-19, this study ignored some factors such as the time of listing that may represent the development of enterprises to a certain extent and are closely related to the financing situation. In view of the above analysis, in further in-depth research, it is necessary to increase consideration of variables such as capital structure, capital market development, and information disclosure norms, eliminate more relevant interference, and clearly show the relationship between sustainable development, carbon information disclosure, and debt financing costs. In addition, predictions of fixed asset ratios and total assets turnover ratios as well as share proportions of the top five shareholders have inspired us to further study the relationship between corporate operations (capital efficiency and asset efficiency, etc.), corporate governance, and debt cost.

**Author Contributions:** Conceptualization, G.W. and X.L.; methodology, J.S. (Jianfei Shen) and E.D.; software, E.D.; validation, X.Z.; formal analysis, E.D.; investigation, J.S. (Jiaxin Shao); resources, J.S. (Jiaxin Shao); data curation, J.S. (Jiaxin Shao) and J.L.; writing—original draft preparation, E.D. and X.Z.; writing—review and editing, J.S. (Jianfei Shen); visualization, E.D.; supervision, G.W. and X.L.; project administration, J.S. (Jianfei Shen); funding acquisition, J.S. (Jianfei Shen). All authors have read and agreed to the published version of the manuscript.

**Funding:** This research received no external funding.

**Institutional Review Board Statement:** Not applicable.

**Informed Consent Statement:** Not applicable.

**Data Availability Statement:** Not applicable.

**Conflicts of Interest:** The authors declare no conflict of interest.

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
