# Peer review of "Corporate Carbon Information Disclosure and Financing Costs: The Moderating Effect of Sustainable Development"

_sustainability, doi:10.3390/su14159159_

Round 1

Reviewer 1 Report

Reviewer Comments on sustainability-1793167

The paper has a potential contribution on the effects of carbon disclosures on the financing costs of firms in 26 industries in China for the period 2009-2021. A number of clarifications are needed.

1) The hypotheses are written in one-sided form while the test results seem to be based on two-tailed tests. Please report the one-sided test significance levels.

2) Lag selection procedure for the dynamics panel is not explained well. Why are four lags selected? In addition, some overfitting and underfitting exercise should be done as a robustness check.

3) The models do not include much residual diagnostics or an analysis of the explanatory power. Please report pseudo-R-square values for the models. A Wald test alone is not sufficient.

4) Do the results change over the sample period? Please report goodness-of-fit analysis by years. The same applies for the industries.  Industry-wise analysis of the results is needed. 

5) Linked to point 4, are there influential observations, years, industries, firms? Please report the analysis results.

Reviewer 2 Report

The authors aim to explores the relationship between corporate  sustainable development, carbon information disclosure and financing costs, taking Chinese listed  companies from 2009 to 2021 as research sample. The following research questions have been raised: 1. Will the sustainable development of enterprises reduce the cost of corporate financing? 2 Can corporate carbon information disclosure reduce the cost of corporate financing? 3 Do carbon information disclosure and sustainable development have a synergistic effect on the cost of corporate financing? I think the authors have answer these questions.

The article, however, needs some further improvements in order to be aligned with the aims, scope and standards of the journal: Sustainability. I have listed some specific comments that might help the authors further enhance the manuscript's quality.

1.       Abstract: the abstract needs revisions

The abstract should be a total of about 200 words maximum. The abstract should be a single paragraph and should follow the style of structured abstracts, but without headings: 1) Background: Place the question addressed in a broad context and highlight the purpose of the study; 2) Methods: Describe briefly the main methods or treatments applied. Include any relevant preregistration numbers, and species and strains of any animals used. 3) Results: Summarize the article's main findings; and 4) Conclusion: Indicate the main conclusions or interpretations. The abstract should be an objective representation of the article: it must not contain results which are not presented and substantiated in the main text and should not exaggerate the main conclusions.  The authors need to clarify the background and method of the article.

2.       Research originality.

Please clarify how is your article adds new knowledge to the body of knowledge that already exists in a research area.

3.       Methodology

The authors need to offer a concrete research framework. A diagram might be helpful.

4.       Conclusion

The authors need to draft a conclusion section. The conclusion of the article should be clear and concise and try to address the question and explain how you have met the research objective raised in the introduction. 

Round 2

Reviewer 1 Report

Dear Authors: Thanks for revising your work. We still need a meausure of goodness-of-fit or explanatory power of the model. The explanatory power of the other models is rather low. The Wald test for the FGLS only shows whether the model is overall statistically meaningful (at a given significance level) without telling how tight the fit is. It is true that the FGLS is a transformed model and whether there is a good or bad fit for that model is not as important. Still, I would suggest calculating RMSE or correlation of actual and fitted values and comparing those values to the ones obtained from the other models. Is this possible? Another issue is the formatting of the Tables. They need to be improved (especially in Table 4). Landscape orientation can be used when needed.

Reviewer 2 Report

The authors have addressed my queries and I am happy with the revised version of the paper. 
